# A Bidirectional Non-Coding RNA Promoter Mediates Long-Range Gene Expression Regulation

**DOI:** 10.3390/genes15050549

**Published:** 2024-04-25

**Authors:** Carlos Alberto Peralta-Alvarez, Hober Nelson Núñez-Martínez, Ángel Josué Cerecedo-Castillo, Augusto César Poot-Hernández, Gustavo Tapia-Urzúa, Sylvia Garza-Manero, Georgina Guerrero, Félix Recillas-Targa

**Affiliations:** 1Instituto de Fisiología Celular, Departaménto de Genética Molecular, Universidad Nacional Autónoma de México, Mexico City 04510, Mexico; cperalta@ifc.unam.mx (C.A.P.-A.); hnunez@ifc.unam.mx (H.N.N.-M.);; 2Instituto de Fisiología Celular, Unidad de Bioinformática y Manejo de la Información, Universidad Nacional Autónoma de México, Mexico City 04510, Mexico

**Keywords:** bidirectional promoter, cis-regulatory element, long non-coding RNA, transcription regulation, gene structure

## Abstract

Recent evidence suggests that human gene promoters display gene expression regulatory mechanisms beyond the typical single gene local transcription modulation. In mammalian genomes, genes with an associated bidirectional promoter are abundant; bidirectional promoter architecture serves as a regulatory hub for a gene pair expression. However, it has been suggested that its contribution to transcriptional regulation might exceed local transcription initiation modulation. Despite their abundance, the functional consequences of bidirectional promoter architecture remain largely unexplored. This work studies the long-range gene expression regulatory role of a long non-coding RNA gene promoter using chromosome conformation capture methods. We found that this particular bidirectional promoter contributes to distal gene expression regulation in a target-specific manner by establishing promoter–promoter interactions. In particular, we validated that the promoter–promoter interactions of this regulatory element with the promoter of distal gene *BBX* contribute to modulating the transcription rate of this gene; removing the bidirectional promoter from its genomic context leads to a rearrangement of *BBX* promoter–enhancer interactions and to increased gene expression. Moreover, long-range regulatory functionality is not directly dependent on its associated non-coding gene pair expression levels.

## 1. Introduction

Cis-regulatory elements (CREs), such as promoters and enhancers, have been classified according to the following: their capacity to modulate gene expression in a local or distal context, their position regarding a gene transcription start site (TSS), and, more recently, their epigenetic signature [1,2,3]. Nevertheless, the idea that promoters and enhancers constitute two different, discrete, and exclusive categories of regulatory elements has recently been challenged by evidence that suggests that their epigenetic or functional features cannot easily distinguish such regulatory sequences [1]. Both enhancers and promoters contain transcription initiation sites and share architectural features such as the presence of core promoter elements and an abundance of transcription factor binding sites (TFBS) [4,5,6]. Genome-wide, it is typical that the same regions present promoter-like and enhancer-like activity [7]. Moreover, distal enhancer-like regulation has been reported to be carried on by promoters independently or even in opposition to the biological role of their locally regulated gene transcripts [8,9,10,11]. There are many unanswered questions regarding the biological determinants of enhancer or promoter activity and how generalized it is that a single regulatory element can participate in promoter- and enhancer-like regulation [12].

An apparent requisite for enhancer-like gene expression regulation is the need for balanced bidirectional transcriptional activity originating from the regulatory element [12,13]. This requirement is fulfilled by many human gene promoters known as bidirectional promoters, which can simultaneously control the transcription rate of a closely interspaced divergent gene pair [14,15,16,17]. On the FANTOM CAT human genome annotation data set, up to 16% of coding genes and 11% of long non-coding RNA (lncRNA) genes can be found on a divergent arrangement within 1000 base pairs of another coding or lncRNA gene [18] (Appendix A). Human protein-coding bidirectional promoter-regulated gene pairs (BPRGPs) are frequently co-expressed and functionally involved in similar biological processes; the expression of non-coding/protein-coding BPRGPs has also been observed, and in some cases, non-coding genes of a BPRGP are involved in modulating their protein-coding gene partner [19,20,21].

There are many questions about how bidirectional promoter architecture influences its associated gene pair expression and whether it can contribute to regulation in an enhancer-like manner [22].

Here, we report an example of a lncRNA bidirectional promoter with distal regulatory action over gene expression in its topologically associating domain (TAD). The studied bidirectional promoter is regulating the expression of the divergently arranged lncRNA genes *LINC00882* and *DUBR*, which have been reported to be expressed aberrantly on different cancer phenotypes and associated with a cell proliferation phenotype [23,24]. This regulatory region shares features with most protein-coding gene pair-associated bidirectional promoters such as the size of the intergenic region and being enriched with cytosine and guanine nucleotides; this sequence composition is, however, atypical for bidirectional promoters associated with lncRNA gene pairs [25]. Our research demonstrated that this region acts as a bidirectional promoter while it is able to modulate distal *BBX* gene expression by preventing promoter–enhancer interactions via a repressive promoter–promoter chromatin interaction. 

## 2. Materials and Methods

### 2.1. Cell Culture

K562 cells were grown in ISCOVE medium (GIBCO, Waltham, MA, USA) supplemented with 10% fetal bovine serum (FBS) (Biowest, Nuaillé, France) and 1X penicillin-streptomycin (Biowest). MCF-7, MCF10A, and HEK-293T cells were cultivated in DMEM (Biowest) medium containing 10% FBS with 10% fetal bovine serum (GIBCO) and 1X penicillin-streptomycin (Biowest). Culture conditions were 37 °C and 5% CO_2_.

### 2.2. Luciferase Reporter Assay

Putative *LINC00882/DUBR* promoter regions were amplificated utilizing oligonucleotides listed in Appendix A. Oligonucleotides (Sigma, St. Louis, MI, USA) were designed to include: 5‘-Two-nucleotide overhang + BglII motif + DNA specific complementary sequence -3‘. Digested and dephosphorylated amplicons were cloned into pGL3-Basic or pGL3-Promoter luciferase vector (Promega, Madison, MI, USA) previously linearized with BglII enzyme (NEB, Ipswich, MA, USA). Insertion and directionality were screened using double enzymatic digestion. For luciferase activity quantitation, 100,000 MCF-7, MCF10A, or K562 cells were seeded on 6-well plates in triplicate for each tested plasmid, then cotransfected with Renilla luciferase vector and pGL3 constructions using Lipofectamine 2000 (Invitrogen, Waltham, MA, USA) according to the manufacturer instructions. Luciferase activity was measured 24 h after transfection using the Dual-Luciferase Reporter Assay kit (Promega) on a Luminometer TD-20 (Turner Designs, San Jose, CA, USA). Internal normalization and relative luciferase activity were performed according to pGL3 vector manufacturer instructions [26]. Plots and statistical analysis were performed using R (version 4.2.3) and ggplot2 library [27] (version 4.2.3).

### 2.3. Lentiviral Production and Cell Infection

For lentiviral production, 1,000,000 HEK-293T cells were plated on 10 mL of complete media containing 0.3 M MgCl_2_, 2 × HEBS (280 mM NaCl, 10 mM KCl, 1.5 mM Na_2_HPO_4_, 12 mM D glucose and 50 mM HEPES [pH 7.05]) and a mix of plasmids as follows: 10 μg of the vector of interest, 3 μg of pMD2.G, and 6 μg of psPAX2. After an overnight period, fresh complete media was used to replace transfection media. The supernatant containing the virus was harvested 48 h post-transfection and centrifuged for 90 min at 27,000 rpm at 4 °C. The pellet containing the viral particles was eluted with 1 × PBS overnight at 4 °C. This lentiviral supernatant was aliquoted and stored at −80 C. We plated 1 × 10^5^ K562 cells per well in 6-well plates, which were infected with 1 mL of lentiviral supernatant and 2 mL of media supplemented with 8 μg/mL polybrene (Sigma). Lentiviral media was replaced by fresh complete media supplemented with a selection agent (puromycin) 24 h after infection. Cells were maintained with the selection agent for ten to fourteen days before the experimental analysis.

### 2.4. CRISPR-Cas9 Plasmid Construction

Single guide RNAs (sgRNAs) were designed with the CRISPOR web tool (http://crispor.tefor.net/ (accessed on 20 January 2018)) [28] using the hg19 genome annotation. Single guide RNAs with high specificity scores (>70), as recommended by the web tool, were selected (Sigma). For genetic deletions with CRISPR-Cas9, single guide RNAs used to generate T1, T2, B1, B2, and Δ *DUBR* mutants were cloned into lentiCRISPRv2 (donated by Feng Zhang, Addgene plasmid # 52961). Cloning was carried out as previously described [24]. All the primers and single guide RNA oligonucleotides used in this study are listed in Appendix A.

### 2.5. CRISPR-Cas9 Assay

CRISPR-Cas9-directed deletion utilized single guide RNAs cloned into lentiCRISPRv2 vector. Upon lentiviral particle generation, 10,000 K562 cells per well were infected in 6-well plates with 1 mL of lentiviral supernatant in 2 mL of media supplemented with 8 μg polybrene (Sigma). Lentiviral media was replaced by fresh complete media supplemented with a selection agent (puromycin) 24 h after infection. Cells were maintained with the selection agent. After ten to fourteen days of selection, a cell pellet was used for DNA extraction by phenol-chloroform. Desired deletions were analyzed by PCR utilizing oligonucleotides listed in Appendix A. To obtain monoclonal mutant cell cultures, we took 10,000 pool cells and serially diluted them in a 96-well plate containing 100 µL of complete ISCOVE medium per well. After fourteen days, single clones were identified by microscopy and expanded for subsequent genotypification by PCR. Deletion of each mutant cell clone was further characterized by cloning PCR fragments obtained from genotypification into pGEM-T Easy (Promega) and confirmed by Sanger sequencing.

### 2.6. Cell Proliferation Assays

For B1 and B2 cell clones and their respective K562 wild-type and empty vector controls, cell proliferation was estimated by Trypan blue staining on a Neubauer chamber. In brief, 50,000 cells were seeded in a 96-well plate with three replicates for each day (4 days, 3 replicates) for each measured cell line. Every 24 h, three wells were vigorously pipetted, and an aliquot of cell culture was taken and diluted with Trypan blue. Four fields of the Neubauer chamber were considered, and every well was quantified twice; live cells per milliliter (unstained cells × 10,000/4 × dilution factor) and live cell percentage (unstained cells/stained + unstained cells × 100) were estimated, and the average of each well measurement was considered for the report. The *DUBR* mutant clone proliferation assay was performed using an MTT-based method: we seeded 50,000 cells per well in a 96-well plate. Cell proliferation was measured by Cell Proliferation Kit (MTT) (Roche) for 4 days according to the manufacturer protocol. All statistical analyses were performed using R (version 4.2.3) base libraries.

### 2.7. RNA Isolation and RT-qPCR

Total RNA was isolated using TRIzol Reagent (Invitrogen) according to the manufacturer protocol. Isolated RNA was used directly to determine gene abundance by KAPA SYBR FAST One Step kit (KAPA Byosystems, Wilmington, MA, USA) using the StepOne Real-Time PCR System. Oligos targeting POLR3A transcript were used as an internal control. RT-qPCR data was analyzed by the ΔΔCt method [29]. Significance in gene expression was determined by Student’s t-test using the R programming language (version 4.2.3) base libraries. All the primers used in this study are listed in Appendix A.

### 2.8. RNA Sequencing (RNA-Seq) and Data Analysis

Total RNA was isolated from wild-type and BBQ mutants (B1 and B2) by triplicate using TRIzol reagent (Thermofisher, Waltham, MA, USA) as described above. Three independent libraries per condition were sequenced in an Illumina HiSeq 4000 platform as paired-end (25 × 10^6^ reads per sample) 150 bp reads. Sequencing reads were aligned to the human genome assembly hg19 (GRCh37) using Bowtie2 [30] (version 2.4.5) with default parameters. Gene counts were calculated using FeatureCounts [31] (version 2.0.6) with parameters “-p”, “-t exon”, and “-g gene_id”. Differential expression was estimated using DESeq2 [32] package (version 4.2.2), arbitrary thresholds for significance were established at absolute log2(Fold change) > 1.5 and adjusted *p*-value < 0.0001, ggplot2 [27] (version 4.2.3) library was utilized for additional plotting. Differentially expressed genes are reported for all conditions in Appendix A. All reads have been deposited at the National Center for Biotechnology Information and can be accessed in the GEO database under accession number GSE264638.

### 2.9. Site-Directed Mutagenesis

Directed mutation over the MYC binding site [33] in the BBQ element was performed by PCR amplification of existing pGL3-BBQ plasmids utilizing oligonucleotides with the desired substitutions (Appendix A); amplification products were then digested with DpnI (NEB), *E. coli* TOP10 competent cells were transformed and grown, and mutations were confirmed by Sanger sequencing.

### 2.10. Chromatin Immunoprecipitation (ChIP)

ChIP was performed based on previous methods [34,35], with some modifications. Briefly, PBS-washed cells were fixed for 10 min in PBS with 1% formaldehyde at a density of 5 × 10^6^ cells/mL. Crosslinking was quenched for 5 min with 0.125 M glycine. Cells were collected and washed with PBS supplemented with 1× PMSF. The pellet was lysed (10 mM Tris-HCl pH 7.5, 10 mM NaCl, 0.3% NP-40, 1× cOmplete (Roche)) and incubated for 30 min. Nuclei were isolated by centrifugation and resuspended in Nuclear Lysis Buffer (50 mM Tris-HCl pH 7.5, 10 mM EDTA, 1% SDS, 1× cOmplete). Chromatin was fragmented in a Bioruptor (Diagenode, Liege, Belgium) for 300 s (amplitude 35%, 9.9 s pulse/9.9 s pause). A volume corresponding to 50 µg of sonicated chromatin was diluted 1:5 with dilution buffer for each sample (1% Triton X-100, 2 mM EDTA, 20 mM Tris-HCl, 150 mM NaCl, and PIK). Chromatin was precleared with 50 µL of blocked protein G/A beads for 2 h (1% of the volume from each sample was taken as input for the ChIP after pre-cleaning and stored at 4 °C) and incubated overnight at 4 °C with 2 µg of antibody c-Myc (9E10): sc-40X (Santa Cruz Biotechnology, Lot #B2521). ChIPs were carried by using 30 µL of blocked protein G/A beads for 2 h at 4 °C. Beads were washed as follows: four times with 1 mL of Wash Buffer I (0.1% SDS, 1% Triton X-100, 2 mM EDTA, 20 mM Tris-HCl, 150 mM NaCl and PIK), once with 1 mL of Wash Buffer II (0.1% SDS, 1% Triton X-100, 2 mM EDTA, 20 mM Tris-HCl, 500 mM NaCl and PIK) and once with a solution of 10 mM Tris-HCl pH 8. Chromatin was eluted at room temperature in 300 µL of Elution Buffer (1% SDS and 100 mM NaHCO_3_). We added 11 µL of Decrosslink Buffer (200 mM Tris-HCl, 400 mM NaCl, 10 mM EDTA) supplemented with RNase A (Ambion, Austin, TX, USA) and Proteinase K (NEB) at 37 °C and 65 °C, respectively. DNA was purified by Phenol:Chloroform:Isoamyl alcohol (Invitrogen) extraction method. ChIP-qPCR reactions contained 1 µL of ChIP DNA or 1% input and were performed using iTaq Universal SYBR Green Supermix (Bio-Rad, Hercules, CA, USA).

### 2.11. Chromosome Conformation Capture Assay (4C-Seq) and Data Analysis

Primers to obtain 4C viewpoints and 4C-seq libraries were performed essentially as described by Krijger and collaborators [36]. Candidate regions to be used as 4C viewpoints were those that were in the closest proximity to a regulatory element of interest (BBQ element or *BBX* TSS) with a suitable oligonucleotide design according the “Primer designer for 4C viewpoints” web tool https://mnlab.uchicago.edu/4Cpd/ (accessed on 3 August 2020). For the library preparation process: ten million cells were crosslinked with 2% formaldehyde for 10 min, followed by 5-min quenching with glycine. Cells were washed once with PBS, and then incubated for 20 min in lysis buffer (10 mM Tris-HCl pH 8, 10 mM NaCl, 0.2% Igepal) with 1x protease inhibitors. First, three rounds of digestion were carried out with DpnII (NEB R0543S) at 37 °C in a thermomixer at 500 rpm (100U for 4 h, 100U overnight, and 100U for another 4 h); the restriction enzyme was inactivated by heating to 62 °C for 20 min while shaking at 500 rpm. Next, the first ligation was performed with 2000U of T4 DNA ligase (NEB M0202L) at 16 °C in 7 mL of Milli-Q water overnight. Samples were then phenol-chloroform extracted and ethanol precipitated, and the second digestion was performed overnight with 50U of NlaIII (NEB R0125S) or HindIII (NEB R0104S). The second ligation was performed in 5 mL total with 3000 units of T4 DNA ligase (NEB M0202L). The second ligation material was purified with SPRI beads, and it was quantified with a Qubit dsDNA Assay Kit. The first round of PCR amplification with viewpoint-specific primers (Appendix A) was performed with 4 × 50 µL PCR reactions with 200 ng of 4C template using 16 PCR cycles and the Phusion polymerase (NEB M0530L). A second round of PCR with universal primers containing Illumina adapters was performed, and the material was purified with SPRI beads. Two independent libraries per condition were sequenced in an Illumina HiSeq 4000 platform as single-end 150 bp reads. After adapter and quality-based trimming, reads were aligned to the hg19 human genome assembly utilizing pipe4C (version 1.1.6). To call valid chromatin interactions, we adapted Otsu’s [37] method for image thresholding to separate the signal from the noise. In brief, we utilized the “Chromatin combined segmentation” dataset (https://genome.ucsc.edu/cgi-bin/hgTrackUidb=hg19&g=wgEncodeAwgSegmentation (accessed on 1 October 2023)) for K562 cells from the UCSC Table Browser as reference annotation to obtain a count matrix using FeatureCounts [31] (version 2.0.6) with default parameters; for each condition replicate, counts were transformed with a logarithm function, scaled to an 8-bit interval, and a threshold value was obtained by applying the otsu() function. Regions with transformed counts above threshold value were considered valid interactions (Appendix A). To obtain differential interactions upon BBQ mutations, we used count matrices obtained in the previous step as input for DESeq2 [32] (version 4.2.2); arbitrary thresholds for significance were established at absolute log2(Fold change) > 3.5 and adjusted *p*-value < 0.00001, and the ggplot2 [27] (version 4.2.3) library was utilized for data plotting. Differential contacts upon BBQ mutations are listed in Appendix A. All reads have been deposited at the National Center for Biotechnology Information and can be accessed in the GEO database under accession number GSE264663.

### 2.12. Retrieval of Public Data Utilized in This Work

ENCODE’s [38] signal tracks shown in this work were retrieved and plotted using the Integrative Genomics Viewer [39] graphic interface. FANTOM5 data for K562 CAGE [40] and long non-coding RNA annotation [18] were downloaded from the FANTOM project website. (https://fantom.gsc.riken.jp/ (accessed on 1 October 2023)). NCBI hg19 RefSeq annotation, CpG island files, and ENCODE Combined Segmentation (ChromHMM + Segway) dataset from the UCSC Table Browser website (https://genome.ucsc.edu/cgi-bin/hgTables (accessed on 1 October 2023)) [41].

## 3. Results

### 3.1. LINC00882-DUBR Intergenic Region Behaves In Vitro as a Bidirectional Promoter

We first investigated whether a bidirectional promoter regulates the human long non-coding RNA gene pair *LINC00882* and *DUBR*, located at chromosome 3 (q13.12), whose canonical transcription initiation sites are separated by 53 base pairs. We selected a DNA region of ~300 base pairs as a candidate promoter, considering its enrichment on features typically associated with promoters, such as the histone H3K4me3 ChIP-Seq signal, the DNase-seq signal, and CAGE signal focalization (Figure 1a) [42,43]. In order to assess bidirectional promoter activity in the candidate region (from now on, this will be termed BBQ for Bidirectional promoter and *BBX* Quencher due to an associated function that will be described later), we cloned a 271 bp DNA fragment comprising the intergenic region and most of the TFBS and DNase-seq signal into a luciferase reporter vector in a forward and reverse orientation. In addition, we generated constructs with the BBQ region as well the histone H3K4me3 enriched upstream sequences to *LINC00882*, *DUBR*, or both (Figure 1a); for each insert, a forward and reverse orientation vector was obtained. Luciferase activity assays demonstrated that the BBQ fragment is able to initiate transcription of a reporter gene in MCF-7, MCF10A, and K562 cell lines, regardless of BBQ orientation (Figure 1b). Interestingly, no promoter activity was observed on vectors in which cloning direction resulted in upstream regions inserted next to a reporter TSS. However, luciferase activity was detected on all constructs when BBQ was placed next to a reporter TSS (Figure 1b), suggesting that the BBQ element behaves as a bidirectional promoter. Interestingly, flanking regions do not contribute significantly to its activity. To further dissect the bidirectional activity of BBQ, we created reporter plasmids containing DNA fragments of BBQ (Figure 1c). A luciferase activity assay demonstrates that BBQ fragments display some promoter activity in particular central and final segments. However, their capacity to activate luciferase transcription was lower than the full-length BBQ element (Figure 1c). Altogether, these results suggest that bidirectional promoter activity for the lncRNAs LINC00882 and DUBR could be regulated by a region of ~300 nucleotides, whose integrity is linked to their capacity to induce reporter gene transcription but do not seem to receive additional regulatory signaling by their flanking regions; therefore, BBQ behaves as a bidirectional promoter on our tested cell models.

### 3.2. BBQ Monoallelic Deletion Affects Cell Proliferation

Next, we aimed to remove the BBQ element from its genomic context to further evaluate its contribution to the regulation of gene expression. To this end, a CRISPR-Cas9 assay targeting PAM sequences near the BBQ limits was designed (Figure 2a). Using this strategy, we obtained clones that were genotyped by PCR, isolating those with an apparent single amplification band, indicating the deletion of both BBQ alleles (Appendix A). However, none of these putative biallelic mutant clones were able to proliferate or survive after three weeks of clonal expansion. This surprising finding points toward a relevant regulatory role for the BBQ element for cell survival. It is important to note that only monoallelic clones could survive and proliferate, suggesting that at least one target of BBQ regulation is essential for cell viability. Two monoallelic BBQ mutant clones proliferated enough to evaluate their proliferation rate; we termed such clones B1 and B2 (Figure 2a). Monoallelic BBQ mutant clones displayed a significantly slower proliferation rate compared to their wild-type or empty vector transduced counterparts (Figure 2b,c). This altered cell proliferation phenotype can be related to previous reports that associate LINC00882 or DUBR levels with cell duplication rate [23,44].

### 3.3. BBQ Deletion Induces Transcriptional Changes Independently of LINC00882 or DUBR Expression

We performed a transcriptome-wide comparative analysis (RNA-Seq) of B1 and B2 clones to evaluate the effect of BBQ removal on potential regulatory targets. Compared to wild-type control, B1 cells presented 2552 and 2334 genes with significant increased or decreased expression, respectively (absolute log2(fold change) > 1.5 and adjusted *p*-value < 0.0001). Conversely, B2 cells presented 3027 overexpressed genes and 2585 underexpressed genes. Overall, B1 and B2 clones share 3662 differentially expressed genes (Appendix A). Surprisingly, no statistically significant transcript level changes were found on either of the lncRNAs, whose TSS overlaps the BBQ sequence (Figure 3a). Furthermore, LINC00882 showed a nonsignificant increase in the B1 clone. We hypothesize that BBQ was removed in only one allele of the mutant clones; unaffected alleles could be compensating or overcompensating BBQ-associated lncRNA genes. Nevertheless, that would neither explain the observed proliferation phenotype nor the differentially expressed genes shared by both mutant clones.

To address this, we opted to compare the genome-wide expression of mutant clones to a *DUBR*(+/−) CRISPR-Cas9 mutant clone (D1) (Figure 4a). We chose *DUBR* since its transcript was more easily detected on our RNA-Seq assay on a 17:1 ratio compared to LINC00882 (Appendix A). Similarly to BBQ mutants, the *DUBR* mutant presented lower proliferation than wild-type or empty vector controls (Appendix A). Transcript quantification from our RNA-Seq results demonstrated significantly lower levels of DUBR (Appendix A). Principal component analysis of the transcriptomes demonstrated that the BBQ clones differ from wild-type controls as well as from the *DUBR* mutant clones (Figure 3b). In addition, BBQ clones’ expression patterns over BBQ mutant differentially expressed genes (DEGs) were distinct from those observed on the *DUBR* mutant clone; BBQ-*DUBR* shared DEGs constitute only 9.2% of the total BBQ mutant DEGs (Figure 3c,d). Despite such differences, the intersection between *DUBR* and BBQ DEGs demonstrated the enrichment of biological pathways that could be associated with their cell proliferation phenotype (Figure 3d,e).

These results demonstrated that loss of one BBQ allele leads to transcriptional changes that are not dependent on their locally regulated genes and differ from those resulting from *DUBR* knockdown. It is interesting to note that despite transcriptomic differences between BBQ and *DUBR* mutant clones, both experimental conditions are associated with a reduced proliferative phenotype. Consequently, we speculate that BBQ has an alternative regulatory mechanism besides its role as a gene promoter. Considering recent reports of promoters with enhancer-like functions, we considered it relevant to elucidate whether BBQ might regulate genes through chromatin interactions and if such interactions would occur in differentially expressed genes, at least on a topologically associating domain (TAD) level [8,10,34,45]. 

To evaluate a possible long-range regulatory role of BBQ, we searched for published BBQ–gene promoter contacts on hematopoietic lineage cells and contrasted them with our transcriptomic results. Within its TAD, the BBQ element interacts with *CCDC54*, *BBX*, *CD47* promoters, and the *CIP2A/DIZP3* putative bidirectional promoter (Figure 3e) [46,47]. Of those known interactions, *BBX* was differentially overexpressed (log2(fold change) > 1.5 and adjusted *p*-value < 0.0001) in both BBQ mutant clones. However, no transcript level changes occurred on the *DUBR* mutant clone. In addition, the *CIP2A* transcript level increased significantly in the B1 clone; however, in the B2 clone, log2FC was slightly below the significance threshold (1.3 log2FC), and no significant changes were detected on the *DUBR* mutant clone (Figure 3e). These findings suggest a possible long-range regulatory role of BBQ, at least in our cell model, and such regulation would be toward transcriptional repression. Regarding the role of DUBR, it is unclear if the presence of such lncRNA might be essential to activate gene transcription upon BBQ elimination, in particular, because BBQ and *DUBR* mutants share a 119 nucleotide region, including canonical TSS for *DUBR* gene (Figure 4a). 

**Figure 3 genes-15-00549-f003:**
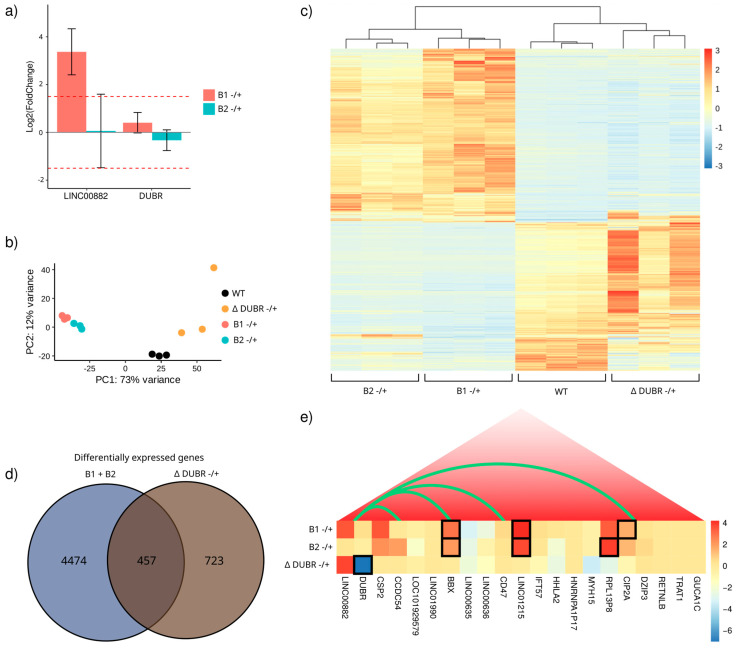
BBQ elimination leads to transcriptional changes independently of LINC00882 or DUBR transcript levels. (**a**) LINC0082 and DUBR expression changes compared to wild-type transcript level expressed as log2(fold change); DESeq2 significance threshold was established at an absolute value of log2(fold change) > 1.5 and adjusted *p*-value < 0.0001. Dotted red lines mark −1.5 and 1.5 log2(Fold change). (**b**) Principal component analysis of DESeq2 normalized counts for all genes in BBQ mutant clones (B1 and B2), *DUBR* mutant clone, and wild-type K562 cells. (**c**) Heatmap of Z-score scaled DESeq2 normalized counts of B1 and B2 differentially expressed genes compared with wild-type K562 cells. All three replicates of B1, B2, wild-type cells, and *DUBR* mutant clone are represented. (**d**) Venn diagram of B1 and B2 differentially expressed genes (B1 + B2 vs. WT) compared with the *DUBR* mutant clone differentially expressed genes (Δ *DUBR* vs. WT). (**e**) Not-to-scale representation of Rao 2014 topologically associating domain genes [47] and their Z-score scaled DESeq2 normalized counts for BBQ mutants and the *DUBR* mutant. Boxes with black borders indicate significant differential expression compared to wild-type cells. Green arcs mark Javierre 2016 [46] capture Hi-C interactions for hematopoietic cells.

**Figure 4 genes-15-00549-f004:**
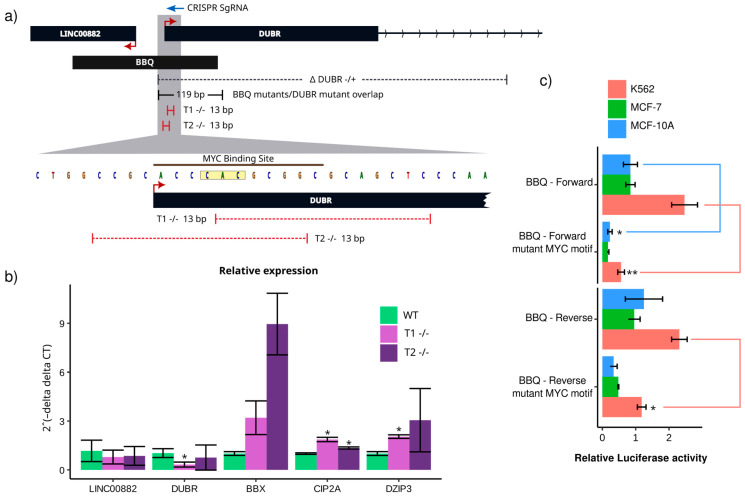
*DUBR* transcription start site mutation is important for bidirectional promoter activity and leads to long-range gene expression changes. (**a**) Top: Representation of the affected genomic region of the BBQ and *DUBR* mutants, sgRNA (blue arrow), and resulting deletions (red lines) of CRISPR-Cas9 mutant clones (T1 and T2). Bottom: MYC transcription binding site and in vitro site-directed mutagenesis replaced nucleotides (yellow box). (**b**) RT-qPCR for LINC0082, DUBR, and distal genes transcripts (n = 6). *t*-test significance of mutant clone gene levels against wild-type control (*p*-value < 0.05 = *). (**c**) Luciferase reporter assay of MYC site mutant constructs compared with wild-type BBQ constructs on K562, MCF-7, and MCF10A. Significant differences between each MYC site mutant construct versus its respective wild-type counterpart were tested using the Mann–Whitney–Wilcoxon test (*p*-value < 0.001 = ** and *p*-value < 0.01 = *).

### 3.4. Canonical DUBR Transcription Start Site Affects Distal Gene Expression

To investigate the shared regions involved in the regulatory participation of BBQ and *DUBR* mutant clones, we directed CRISPR-Cas9 to a PAM sequence next to the *DUBR* canonical TSS (Figure 4a). We obtained mutant clones (T1 and T2) with deletions (13 bp) that overlap canonical *DUBR* TSS (Figure 4a and Appendix A). We did not detect a wild-type sequence over repeated Sanger sequencing of PCR products from T1 or T2 clones; we therefore considered them to be biallelic mutants. Interestingly, just one of our mutant clones had significantly decreased DUBR expression. No mutant clone had substantial changes in LINC00882 expression (Figure 4a). Distal genes with known chromatin contacts with the BBQ locus exhibited higher transcript levels than the wild-type control (Figure 4b). This further supports the role of BBQ as a long-range cis-regulatory element and is in concordance with our luciferase results, as a partial deletion of BBQ does not entirely eliminate its promoter activity (Figure 1d).

The effect of mutations on the *DUBR* TSS at a mechanistic level is uncertain. The lack of significant expression changes in both lncRNAs; DUBR or LINC0082, may be a result of an alternative TSS, which has been observed for the *LINC00882-DUBR* gene pair on the FANTOM5 CAGE dataset [38]. As previously explored, distal gene expression changes upon *DUBR* TSS or BBQ elimination might be caused by the deregulation of functional chromatin interactions. In such cases, our mutations would interfere with an active functional sequence. On ENCODE 3 ChIP-Seq peaks datasets for K562, c-Myc was the unique binding agent to the region with a reported TFBS motif over the region that was removed on our *DUBR* TSS mutant clones [38,48].

Therefore, we explored whether a Myc binding site mutation could affect in vitro BBQ bidirectional promoter activity. Using a site-directed mutagenesis approach, we replaced existing Myc binding site core nucleotides (5’-accCACgcggc-3’ to 5’-accGGTgcggc-3’) and measured in vitro promoter activity as previously described. The mutant version of the reporter vector demonstrated significantly lower promoter activity than its wild-type counterpart in K562 cells, regardless of cloning orientation (Figure 4c). We validated that our T1 and T2 clones presented reduced c-Myc binding compared to wild-type cells, resulting from the elimination of the MYC binding site (Appendix A). We verified that our in vitro mutagenesis and our T1 and T2 clones were not reconstituting a MYC binding site using FIMO from the MEME suite. Our experimental approach allows us to conclude that the sequence overlapping the MYC motif is important to BBQ distal regulatory activity; however, a more comprehensive approach is required to validate that c-Myc is an active participant in BBQ activity. In addition, c-Myc is only one of the possible regulators of this locus, although its binding motifs do not directly overlap with the T1 or T2 mutated region; according to ENCODE [38] ChIP-Seq data, there are fourteen proteins with enrichment peaks that overlap with the BBQ element (Appendix A). In addition, it is also plausible that DUBR or LINC00882 participate in BBQ element distal regulatory activity. 

### 3.5. BBQ Physically Interacts with Differentially Expressed Genes

To investigate how BBQ can influence gene expression via chromatin interactions, we designed a chromosome conformation capture experiment with K562 wild-type, BBQ mutant (B1), and two *DUBR* TSS mutant clones (T1 and T2). We opted for a 4C-Seq approach by considering that known functional interactions overlapping the BBQ element were obtained using a promoter capture enrichment variation of the Hi-C method [46]. To obtain comparable results, we selected an adjacent region to the BBQ element as our 4C viewpoint. This region was chosen as oligonucleotide design directly over the bidirectional promoter was not possible, as well as to avoid any possible artifacts resulting from BBQ partial elimination (Figure 5a and Appendix A). To study chromatin interactions from a functional perspective, we utilized ENCODE’s Combined Segmentation (ChromHMM + Segway) functional annotation of K562 chromatin as a reference annotation to call valid interactions and further comparative analysis. This dataset divides chromosome 3 into 63,963 regions, according to their combinatorial histone post-translational modifications, chromatin accessibility, transcription factor binding, and transcriptional activity and assigns them a functional category based on this chromatin landscape (Figure 5a) [2,3,38,49].

On K562 wild-type cells, 3654 valid contacts were detected for the BBQ element, including all previously described BBQ chromatin contacts [46]. In addition, on the BBQ topologically associating domain, we found an additional 11 (out of 21) genes that interact with BBQ on a genome segment near their TSS (+/− 3 kb) (Appendix A). Interestingly, on a scope restricted to the BBQ TAD context, three out of the four genes that previously showed differential expression in at least one BBQ mutant clone presented a TSS overlapping valid interaction with the BBQ viewpoint sequence on K562 wild-type cells (*BBX*, *LINC01215*, and *CIP2A*) (Figure 5b) (Appendix A). The remaining overexpressed transcript upon BBQ elimination corresponds to a pseudogene located entirely within a genome segment marked with a repressive chromatin signature, which acquired valid BBQ interactions on all three mutant cell clones (Appendix A).

Next, to evaluate BBQ element participation in maintaining BBQ-promoter chromatin interactions, we compared read counts over ENCODE functional annotation segments using DESeq2; we established an arbitrary threshold of 3.5 log2(fold change) (~ 11 fold difference to K562 wild-type mean) and an adjusted *p*-value < 0.00001. With this approach, we found—on at least two out of three mutant clones—significant differential BBQ interactions on all three genes and pseudogene that changed their expression upon full or partial deletion of the BBQ element (Figure 5b). For *BBX* and *LINC01215* genes, BBQ presented a reduced interaction to ENCODE’s functional annotation segments with a promoter-like epigenetic signature overlapping their TSSs (Figure 5c,e and Appendix A). The *CIP2A-DZIP3* gene pair is in a closely interspaced divergent arrangement, for which interactions with the BBQ element did not significantly change; instead, a *DZIP3* intronic region with functional annotation that corresponds to repressed chromatin, shows decreased BBQ interactions upon total or partial BBQ deletion (Figure 5c and Appendix A). The entire *RPL13P8* pseudogene sequence overlaps with a genome segment annotated as repressed chromatin and was the only example of an increase in BBQ interactions. However, those interactions were not restricted to TSS vicinity (Figure 5c,d). Finally, differential interactions upon BBQ deletions were observed on the TSS of *CD47* and *IFT57*, which are genes that did not present differential expression in our RNA-Seq results (Figure 5c). It is interesting to note that all differentially expressed genes upon BBQ mutation presented differential interaction frequency with the BBQ element; this supports the idea that BBQ is a long-range regulatory element whose interactions might have a repressive role (*BBX*, *CIP2A-DZIP3*, and *LINC01215*), an activating role (*RPL13P8*), or neutral effect on gene expression (*CD47* or *IFT57*). All of this is suggestive of target-specific activity.

To test whether the BBQ element has an intrinsic long-range effect on gene expression, we designed a reporter assay experiment with BBQ cloned not as a promoter but as an enhancer (Appendix A). We observed that BBQ does not significantly modify the SV40 promoter potential to initiate luciferase transcription (Appendix A). This might be interpreted as BBQ lacking intrinsic regulatory activity. However, it could be also interpreted as BBQ requiring another molecule, such as one or both of the following: DUBR or LINC0082, for example. Furthermore, this could explain the lack of differential expression of BBQ putative gene targets on the DUBR knockdown model (Figure 3e). Finally, another possibility is that the BBQ effect on contacted regions is highly specific, which would be supported by observed differential BBQ interactions with no evident gene expression change.

Together, these results suggest that the BBQ element might have a target-specific regulatory effect on gene expression through the formation of functional chromatin interactions with gene promoters; it is unclear, however, if such regulation is executed by BBQ itself or if it might require the participation of other regulatory effectors.

### 3.6. BBQ Acts as a Quencher of BBX Expression and Modulates Regulatory Chromatin Interactions

We next explored how a putative target for BBQ-mediated regulation would change chromatin interactions upon BBQ partial and total elimination. A 4C-Seq approach was followed, this time with a viewpoint set on *BBX* TSS proximity (Figure 6a and Appendix A). Differential interaction calling demonstrated significant augmented *BBX* viewpoint interactions toward two genomic segments annotated as enhancers that are located at 4.7 kb (distal enhancer) and 133 bp (proximal enhancer) to the BBQ element on three and two mutant clones, respectively (Figure 5a and Figure 6b). An exploration of *BBX* promoter-promoter interactions in its TAD context revealed significant interaction changes toward genes that were observed to be differentially expressed upon BBQ deletions. Furthermore, *LINC01215* TSS demonstrated a consistently reduced interaction frequency to *BBX* viewpoint, and biallelic partial mutations of BBQ displayed reduced interactions toward the *RPL13P8* pseudogene region. Finally, B1 and T1 clones demonstrated increased interactions directly toward the *CIP2A-DZIP3* promoter (Figure 6c). These results contribute to the idea that the BBQ element is not a direct transcription regulation effector but a facilitator of regulatory chromatin interactions, and its deletions lead to a specific TAD-wise rearrangement.

Altogether, our results indicate the following: The interaction of the BBQ element with *BBX* promoter has a repressive *BBX* expression effect; the *DUBR* TSS region appears to play a crucial role in orchestrating BBQ–*BBX* repressive interaction; deletion of BBQ or its *DUBR* TSS section leads to an increase in *BBX* expression and appears to promote *BBX* promoter–enhancer interactions (Figure 6d). In addition, shared *BBX* viewpoint and BBQ differential interactions hint at a more complex network of specific regulatory chromatin contacts; in particular, because *BBX* viewpoint and BBQ interaction changes a gene target, they do not always follow the same enrichment directionality, nor do they occur over the same genomic regions.

## 4. Discussion

In the present work, we studied the functional nature of the BBQ regulatory element, which is a member of one of the least studied groups of regulatory sequences: bidirectional promoters of two non-coding genes. In general terms, bidirectional promoters are more commonly observed in mammals [14,50]. It has been proposed that bidirectional promoters can simultaneously regulate two genes that often participate in similar biological processes [21,51]. Mechanistically, bidirectional promoters are more efficient at RNA Pol II recruiting; however, the functional significance of this is unknown [17]. Stable divergent transcription from bidirectional promoters is a feature that resembles transcription from active enhancers, which has led to the hypothesis that bidirectional promoters might participate in distal gene expression regulation [22,52,53].

Here we demonstrated the participation of BBQ in distal gene expression regulation; however, it is interesting to note that chromatin signatures associated with enhancer activity are not present in the BBQ sequence; in particular, because most long non-coding RNA promoters possess enhancer-like marks [18]. Instead, BBQ displays a promoter-like chromatin landscape with elevated CpG content, which, we observed, is uncommon for non-coding BPRGPs (Appendix A). In summary, the BBQ element does not constitute an example of a typical promoter by its bidirectional nature and also differs from most long non-coding RNA promoters and most long non-coding RNA bidirectional promoters by its sequence and chromatin features. Therefore, we believe that the extrapolation of our findings to a larger group of regulatory sequences should be carried out carefully.

Our findings include that the BBQ element behaves as a typical bidirectional promoter under the classical approach of luciferase activity assays [16]. Removal of BBQ out of its locus context with our CRISPR-Cas9 strategies induced an evident decrement in cell proliferation rate and survival, resulting in the loss of most mutant clones. Instead, surviving clones required an extended period for their expansion up to the point where further tests could be performed. We therefore speculate that the transcriptional state of DUBR and LINC0082, genome-wide transcriptional changes, and three-dimensional reorganization of chromosome 3 constitutes the added effect of BBQ elimination along with long-term cell adaptation changes. We hypothesize that known mechanisms such as allele dose compensation and the use of alternative TSS sites might explain the lack of DUBR and LINC0082 knockdown upon total (−/−) and partial (+/−) BBQ removal. BBQ monoallelic mutant clones presented a proliferative phenotype similar to our *DUBR* knockdown mutant clone; this is interesting, as gene expression profiles on those conditions were distinguishable both at a genome and a TAD level. One possibility is that this locus is modulating regulatory pathways related to the observed phenotype under different mechanisms; one by the action of the genes directly associated with the BBQ element and another by allowing functional chromatin interaction changes of nearby gene promoters such as *BBX* to regulatory elements. We suggest that the specific combination of TFBS in the BBQ sequence might be related to its capacity to modulate cell proliferation-related genes. 

It is currently believed that functional regulatory chromatin interactions are restricted by topologically associating domain boundaries [45,54]. As a result, we considered that to evaluate the possible promoter-like regulatory role of BBQ, we should focus on the chromatin interaction landscape within its TAD. We focused our attention on BBQ-promoter interactions rather than interactions with other regulatory elements in an attempt to capture regulatory effects directly mediated by the BBQ element.

Our chromosome conformation capture assays demonstrated that BBQ displays multiple promoter–promoter interactions along its TAD. Moreover, in this TAD context, significant gene expression changes always coincided with a significant change in BBQ-promoter interaction frequency; this suggests that BBQ–promoter contacts might encompass a regulatory nature. This idea was supported when chromatin contacts were captured using *BBX* TSS as a viewpoint; this gene expression and chromatin interaction were consistently affected by BBQ mutations. In BBQ mutant clones, *BBX* TSS gained contacts with sequences annotated as enhancers. In this scenario, we propose that BBQ is acting as a suppressor of contacts with enhancer elements for and possibly other deregulated genes (Figure 6d). Finally, *BBX* contacts with deregulated genes in BBQ mutant clones were also affected; however, often, such changes neither occurred over the same genome segments nor followed the same direction as those observed from the BBQ viewpoint perspective. Our data might be indicative of a more complex network of promoter–promoter interactions. This idea has been proposed previously as a large number of promoter–promoter interactions have been described genome-wide, and it has been demonstrated that gene expression and promoter–promoter interactions change simultaneously under certain physiological conditions such as circadian rhythms [55,56]. In addition, it has been reported that disruption of certain functional interactions between promoters and enhancers might lead to a rearrangement of interactions with regulatory elements in a specific fashion under a mechanism known as “Enhancer Release and Retargeting” [57]. We think that our observed chromatin interaction changes upon BBQ mutation might be regulated by a similar retargeting process; however, our current methodology does not allow us to demonstrate it. 

Our current approach to the regulatory role of BBQ does not allow us to demonstrate what the input signaling is for its long-range regulatory activity; however, we showed that only a few nucleotides over the *DUBR* TSS are required to change BBQ chromatin interactions. Interestingly, c-Myc is the only transcription factor reported in ENCODE ChIP-Seq datasets that it binds directly to that region. We demonstrated that its binding site mutation affects BBQ promoter activity; furthermore, there are reports in which this transcription factor might contribute to the three-dimensional chromatin organization and has been observed to act as a fundamental transcription factor for other bidirectional promoter regulated genes [58,59]. Another potential player for BBQ functionality is one of its locally regulated genes: the lncRNA DUBR. There were no gene expression changes in BBQ interacting genes upon DUBR knockdown; this might suggest that BBQ interactions required DUBR participation, especially when it was noted that the *DUBR* TSS was also removed in the *DUBR* knockdown clone. The possibility that DUBR plays a role in BBQ distal regulatory activity requires further investigation. In conclusion: BBQ is a regulatory element that combines bidirectional promoter action with a long-range regulatory role in gene expression over genes of its topologically associating domain. It represents another example that promoters and enhancers share many functional similarities. BBQ participation in TAD-wise promoter–promoter interactions contributes to the idea that networks of this type of chromatin interaction adds another layer to the already complex collection of gene expression regulatory mechanisms and to the notion that bidirectional promoter architecture might facilitate long-range gene expression regulation.

## Figures and Tables

**Figure 1 genes-15-00549-f001:**
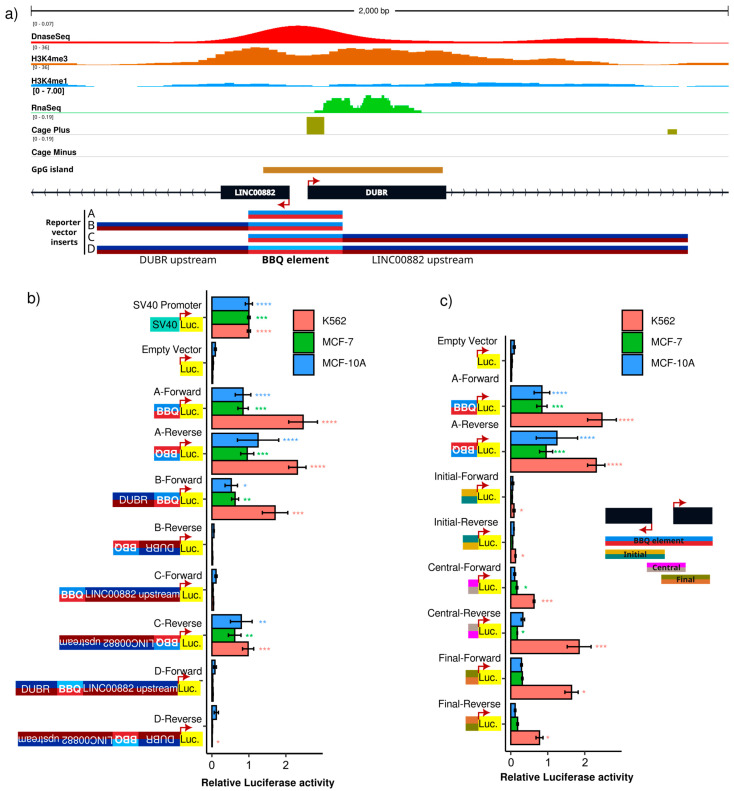
*LINC00882-DUBR* intergenic region displays in vitro bidirectional promoter activity. (**a**) Genomic features around *LINC00882* and *DUBR* transcription start sites, from top to bottom: ENCODE K562 DNase-seq signal, ENCODE K562 H3K4me3 and H3K4me1 ChIP-Seq, ENCODE K562 RNA-Seq signal, FANTOM5 K562 CAGE signal, UCSC Genome Browser CpG islands track, transcription start sites and exons for *LINC00882* and *DUBR* genes, and representation of DNA sequences cloned into pGL3 Luciferase vector. (**b**,**c**) Luciferase activity of cloned sequences expressed relative to pGL3-promoter (SV 40) vector in K562, MCF-7, and MCF10A cell lines (*n* = 9). Significant differences of each construct versus negative control (pGL3 with no promoter) were tested using Mann–Whitney–Wilcoxon test (*p*-value < 0.00001 = ****, *p*-value < 0.0001 = ***, *p*-value < 0.001 = ** and *p*-value < 0.01 = *).

**Figure 2 genes-15-00549-f002:**
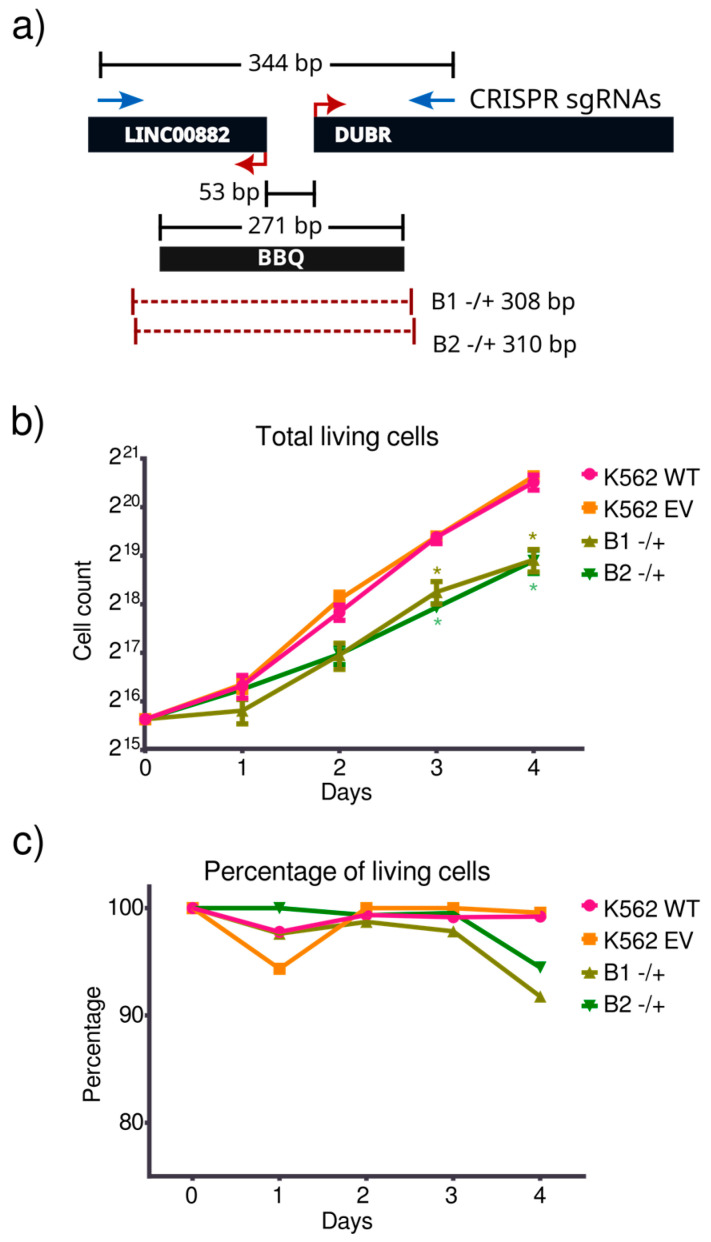
BBQ mutant clones present a proliferation-related phenotype. (**a**) CRISPR-Cas9 assay design. Dual single guide RNA (sgRNA) placement to remove BBQ sequence. Eliminated alleles from both B1 and B2 mutant clones are represented as dotted red lines. (**b**) Total live cell number quantification for B1 and B2 mutant clones compared with K562 wild-type control and CRISPR-Cas9 empty vector cells (n = 3). Significance was measured using Student’s *t*-test (*p*-value < 0.01 = *). (**c**) Percentage of live cells (live cells/total cells ×100) during proliferation challenge for B1 and B2 mutant clones and wild-type and empty vector controls.

**Figure 5 genes-15-00549-f005:**
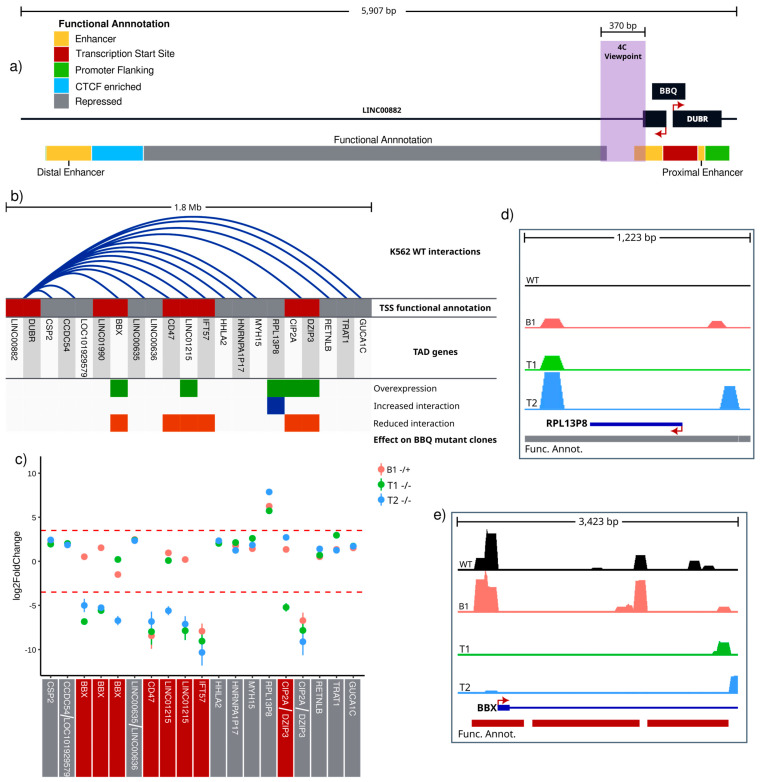
BBQ deletion leads to the loss of BBQ-promoter interactions on differentially expressed genes. (**a**) Map of BBQ locus. The 4C-Seq viewpoint is marked with a purple box. The functional annotation track is based on combined segmentation (ChromHMM + Segway) from ENCODE data for K562 cells; the color key with functional categories is shown in the top left area. (**b**) TAD genes (not-to-scale representation). Blue arcs show valid BBQ–TSS interactions (+/− 3 kb from TSS). Functional annotation of the TSS overlapping genome segments is shown using color code from panel “a”. Genes with significant expression changes measured via RNA-Seq or RT-qPCR on at least one of the BBQ mutant clones (B1, B2, T1, or T2) are marked with green boxes. Genes with TSS genome segments (+/− 3 kb from TSS) with differential interaction with BBQ viewpoint upon BBQ mutation on at least one of the BBQ mutant clones are marked with blue boxes for increased interactions or orange boxes for decreased interactions. (**c**) BBQ viewpoint interaction frequency changes toward TSS overlapping genome segments of TAD genes (+/− 3 kb from TSS), red lines mark −3.5 and 3.5 log2(fold change) arbitrary significance threshold, and all observed changes exceeding fold change threshold also had a *p*-value lower than 0.00001. The color below gene name indicates the functional annotation category according to panel “a” color key table. (**d**,**e**) Scaled and normalized 4C-Seq signal for B1, T1, T2, and WT cells over genome segments near *RPL13P8* (**d**) and *BBX* (**e**) TSS (+/− 3 kb from TSS). The color of genome segments follows panel “a” color key. The signal is shown at a linear scale, and all tracks use the same data range interval.

**Figure 6 genes-15-00549-f006:**
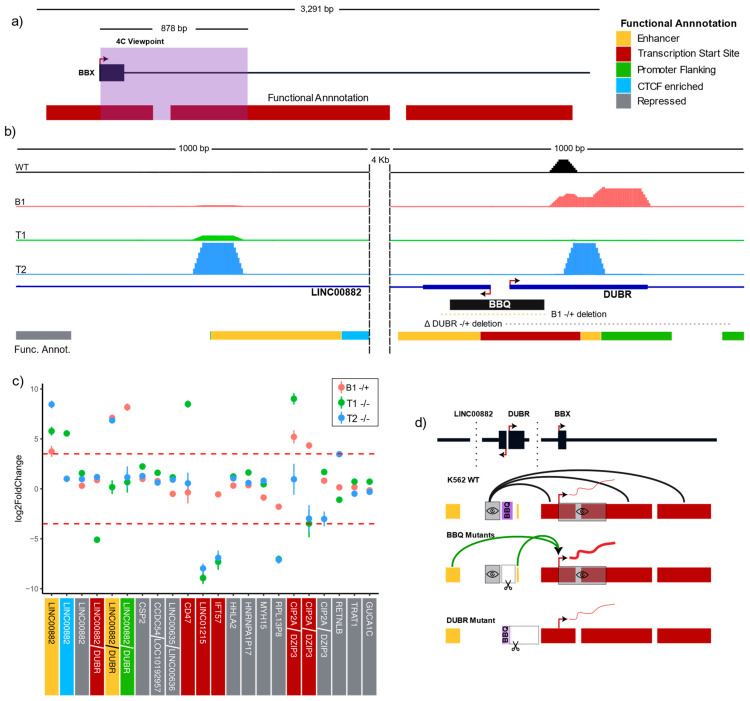
Mutation of the BBQ element is conducive to increased *BBX* promoter-enhancer interactions. (**a**) Map of *BBX* locus. The 4C-Seq viewpoint is marked with a purple box. The functional annotation track is based on combined segmentation (ChromHMM + Segway) from ENCODE data for K562 cells; the color key with functional categories is shown in the top left area. (**b**) Scaled and normalized 4C-Seq signal for B1, T1, T2, and WT cells over genome segments of the BBQ locus (~5 kb); the color of genome segments follows panel “a” color key. The signal is shown at a linear scale, and all tracks use the same data range interval. (**c**) *BBX* viewpoint interaction frequency changes toward TSS overlapping genome segments of TAD genes (+/− 3 kb from TSS and up to 5 kb for the BBQ-associated segments), red lines mark −3.5 and 3.5 log2(Fold change) arbitrary significance threshold, all observed changes exceeding fold change threshold also had a *p*-value lower than 0.00001. The color below the gene name indicates the functional annotation category according to the panel “a” color key table. (**d**) Model for BBQ regulation of *BBX* gene expression. The map shows distal BBQ enhancer, BBQ TSS and associated genome segments, and *BBX* TSS and associated genome segments. Wild-type cells display BBQ interactions toward promoter-like genome segments near *BBX* TSS, Mutant BBQ clones BBQ–*BBX* interactions while increasing BBQ–enhancer interactions, which is associated with increased levels of *BBX* transcript. The *DUBR* mutant also interferes with BBQ integrity but removes the BBQ proximal enhancer and severely affects DUBR expression; in such cases, no significant *BBX* changes are observed.

## Data Availability

All sequencing raw data are available at NCBI repositories under GEO accessions (GSE264663 and GSE264638), All code utilized for this paper is available on the following GitHub repository: https://github.com/cperalta22/BBQ-long-range-expression-regulation.

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
