# Peer review of "A Bidirectional Non-Coding RNA Promoter Mediates Long-Range Gene Expression Regulation"

_genes, 2024, doi:10.3390/genes15050549_

Round 1
Reviewer 1 Report
Comments and Suggestions for Authors
In this study, the authors analyzed changes in gene expression and genomic interactions within the deletion mutant cells of a bidirectional non-coding RNA promoter region. They attempted to elucidate the molecular functions of that promoter using several sequencing techniques, such as RNA-seq and 4C-seq. However, there are some points that are confusing and may lead to misunderstandings. Therefore, the authors should address and further elucidate these points.
Major points:
1. How can we understand some results? B1(+/-) and B2 (+/-) mutants showed no changes in LINC0082 and DUBR expression compared with wildtype control. However, these mutants showed a reduced cell proliferation. DUBR (+/-) mutant also exhibits a low cell proliferation. In B1(+/-) and B2(+/-) mutants, DUBR expression was not altered, while in DUBR (+/-) mutant, DUBR expression was reduced. If so, why B1(+/-) and B2 (+/-) mutants has a reduced cell proliferation phenotype? What genes are related to this cellular phenotype? Have the authors confirmed the expression levels of LINC0082 and DUBR using different approaches (eg, RT-qPCR) in B1(+/-) and B2(+/-) mutants?
2. The authors described that BBQ deletion induced the expression of some genes located within the same TAD. According to a recent paper (PMID: 34040254), enhancer release and retargeting (ERR) can induce some genes within the same TAD once the promoter of a certain gene is deleted. Does the author think that this mechanism is involved in the upregulation of some genes in B1(+/-) and B2 (+/-) mutants?
3. There are differences in expression alterations or genomic interactions between B1(+/-) and B2(+/-) or T1(+/-) and T2(+/-). Is there a possibility of off-targeting by CRISPR/Cas9 during mutant generation?
4. In Figure 5, the authors conducted 4C-seq analysis with one specific viewpoint (a region in LINC00882). Is this known as a specific regulatory region? Why did the authors use that region as a viewpoint? What is the relationship between the reduced interaction of some genes with the viewpoint and the upregulation of those genes?
Minor points:
1. There are some typing errors. Please check the manuscript thoroughly.
2. Figure 2C: Is there no statistical significance on day 4?
3. Did the author provide information on oligonucleotide sequence for site-directed mutagenesis. If not, please include the sequence of oligonucleotides for site-directed mutagenesis in the supplementary table.
Author Response
Thanks for you review report, please see the attachement document.
The authors.

Reviewer 2 Report
Comments and Suggestions for Authors
The article titled " A Bidirectional Non-Coding RNA Promoter Mediates Long-Range Gene Expression Regulation" makes a significant contribution to the literature on Non-Coding RNA.
Evidence suggests that human gene promoters display gene expression regulatory mechanisms beyond the typical single gene local transcription modulation. In mammalian genomes, genes with an associated bidirectional promoter are abundant; bidirectional promoter architecture serves as a regulatory hub for a gene pair expression.
In the present study, the long-range gene expression regulatory role of a long-non-coding RNA gene promoter using chromosome conformation capture methods has been studied. The author found that this particular bidirectional promoter contributes to distal gene expression regulation in a target-specific manner by establishing promoter-promoter interactions. In particular, they validated that bidirectional promoter is contacting multiple gene promoter elements, including BBX promoter, upon loss-of-function assays reduced bidirectional-promoter interactions associated with increased BBX promoter-enhancer and augmented gene expression. Moreover, long-range regulatory functionality is not directly dependent on its associated non-coding gene pair expression levels.
In the present work, the author studied the functional nature of the BBQ regulatory element, which is a member of one of the least studied groups of regulatory sequences: bidirectional promoters of two non-coding genes.
Available reports suggest that bidirectional promoters can simultaneously regulate two genes that often participate in similar biological processes. Mechanistically, bidirectional promoters are more efficient at RNA-Pol II recruiting; however, the functional significance of this is unknown.
This study demonstrated the participation of BBQ on distal gene expression regulation; however, it is interesting to note that chromatin signatures associated with enhancer activity are not present on the BBQ sequence, in particular, because most long-non-coding RNA promoters possess enhancer-like marks. Instead, BBQ displays a promoter-like chromatin landscape with an elevated CpG content, which they observed is uncommon for non-coding BPRGPs. In summary, the BBQ element does not constitute an example of a typical promoter by its bidirectional nature and also differs from most long- non-coding RNA promoters and most long-non-coding RNA bidirectional promoters by its sequence and chromatin features.
Moreover, they demonstrated that its binding site mutation affects BBQ promoter activity; furthermore, there are reports in which this transcription factor might contribute to the three-dimensional chromatin organization and has been observed to act as a fundamental transcription factor for other bidirectional promoter-regulated genes. Another potential player for BBQ functionality is one of its locally regulated genes: the lncRNA DUBR.
In summary, the study underscores the significance of bidirectional promoters like BBQ in facilitating long-range gene expression regulation within topologically associated domains. By shedding light on the functional similarities between promoters and enhancers, the research contributes to our understanding of the complex mechanisms underlying gene expression regulation.
Overall, the manuscript is well-written and presents reasonable conclusions based on the experimental findings.
Author Response

(The authors gave the same response as above.)

Reviewer 3 Report
Comments and Suggestions for Authors
In the manuscript entitled “A bidirectional non-coding RNA promoter mediates long-range gene expression regulation” by Peralta-Alvarez et al., the authors characterized a bidirectional promoter that is in vicinity of two non-coding RNAs – LINC00882 and DUBR. Luciferase reporter assay confirmed the functionality of this region. Using sophisticated chromosome confirmation capture methods, the authors discovered that this regulatory region is involved in distal gene expression regulation in a target-specific manner. More specifically, this region negatively regulates the expression of distally-located BBX promoter. This is an important study and will contribute to the emerging field of long-range gene expression regulation. I overall enjoyed reading the manuscript and have only few comments.
1. I am wondering how did the authors see a significant difference in myc activity just by doing site-directed mutagenesis and not inducing myc expression other way?
2. The authors named the regulatory region ‘BBQ region’ in the beginning when any connection to BBX was yet to be established. The authors might consider naming the region something else.
3. Figure 1B. ‘Insert A’ is missing in the label.
4. There are some typographical errors and grammatical mistakes/incomplete sentences that need to be corrected.
Author Response

(The authors gave the same response as above.)

Reviewer 4 Report
Comments and Suggestions for Authors
In this manuscript, the authors investigate the bidirectionality of the BBQ promoter, demonstrating its capacity to regulate gene expression in distant regions of chromosome 3. Multiple molecular biology techniques and bioinformatic analyses are employed to establish hypotheses regarding the function of the BBQ promoter. Overall, the manuscript is well-written and represents a significant advancement in the study of bidirectional promoters. However, there are several aspects that the authors should address and clarify:
- The penultimate sentence of the abstract needs to be rewritten for better clarity.
- The term "in vitro" should be italicized.
- Several methods and articles are mentioned in the text without specific references, e.g., Krijger PHL and collaborators, Otsu’s method, Javierre 2016...
- In the text, the authors describe the generation of constructs involving the BBQ region and upstream sequences, but the absence of LINC00882 in Figure 1b needs clarification.
- Figure 1c lacks clarity regarding the order of sequences in parentheses and which region is adjacent to the TSS. The authors should revise the figure for better comprehension.
- In Figure 1d, authors should include the complete BBQ element as a control to facilitate comparison of luciferase activity levels among different BBQ fragments.
- Figure S2 requires clarification regarding whether lanes 6 and 7 correspond to mutants B1 and B2, as the PCR results differ significantly from Figure 2a's depiction. In Figure 2a, the dashed red lines indicate that both mutants differ by only 2 base pairs. This discrepancy should be addressed by the authors.
- The expression levels of LINC00882 and DUBR vary greatly with large error bars, making conclusions difficult. Before asserting that BBQ deletion does not alter the expression of these two transcripts, the authors should analyse their expression via qRT-PCR in a larger number of replicates.
- The authors describe B1 mutants as monoallelic but fail to clarify whether other mutants (T1, T2) are monoallelic or biallelic, and why.
- A paragraph explaining the rationale behind 4C-Seq and the authors' decision to use it instead of other methods like Hi-C would be appreciated. Additionally, proper referencing of the method in the text is necessary.
- Supplementary file 2's "BBQ B1 valid interaction" tab is empty.
- In section 3.5, line 18, did the authors actually use 562 wild-type cells? Does this refer to different cell lines? Rewording for clarity is necessary.
- The authors demonstrate that BBQ deletion alters the expression of over 4000 genes but focus on describing interactions for only 21 genes, leaving the alteration of the remaining genes unexplained.
In general, the manuscript is well-written and presents significant advancements in gene expression regulation. Better explanation of techniques and figures could enhance its contribution to the field.
Author Response

(The authors gave the same response as above.)

Reviewer 5 Report
Comments and Suggestions for Authors
The manuscript of Peralta-Álvarez et al. analyzes the role of a sequence between two lncRNAs genes as a bidirectional promoter. The authors performed different analyses to determine the functionality, concluding that it is a regulatory element affecting different genes.
The manuscript is well-written, has an informative introduction, and thoroughly describes the material and methods. The reason for using the specific region selected is missing. It is acceptable that the authors selected it, but why? The lncRNAs selected are related to cancer, but this is not the reason for selecting this section since the aim is to analyze the role of bidirectional promoters. So, it would be helpful for the reader to get any information about the adequacy of the selection in the context of bidirectional promoter analysis.
The results are well described, although sometimes the reader gets lost following the figures. It would be better to reorganize the figures by moving the panel “e” to the bottom. The natural view for the reader is from top to down and from left to right. In the present organization, it is sometimes difficult to understand the meaning of the figure. Have the authors performed any analysis of the BBQ sequence regarding regulatory elements? They have found a site for Myc in a specific region, but it would be helpful to know if other sites are related to regulation.
The discussion fits the results, although it repeats some of them.
Author Response

(The authors gave the same response as above.)

Round 2
Reviewer 1 Report
Comments and Suggestions for Authors
The author has addressed all of the reviewer's points in the revised manuscript, and I have no further comments. Therefore, I recommend the publication of the revised manuscript.
Reviewer 4 Report
Comments and Suggestions for Authors
The authors have addressed many of the suggestions I made in the previous version; therefore, I consider the manuscript suitable for publication. However, I suggest a minor English revision as I have noticed that some sentences are excessively long and could be divided into two.